# FlowHash: Accelerating Audio Search with Balanced Hashing via Normalizing Flow

## Abstract

Nearest neighbor search on context representation vectors is a formidable task due to challenges posed by high dimensionality, scalability issues, and potential noise within query vectors. Our novel approach leverages normalizing flow within a self-supervised learning framework to effectively tackle these challenges, specifically in the context of audio fingerprinting tasks. Audio fingerprinting systems incorporate two key components: audio encoding and indexing. The existing systems consider these components independently, resulting in suboptimal performance. Our approach optimizes the interplay between these components, facilitating the adaptation of vectors to the indexing structure. Additionally, we distribute vectors in the latent $\mathbb{R}^K$ space using normalizing flow, resulting in balanced $K$-bit hash codes. This allows indexing vectors using a balanced hash table, where vectors are uniformly distributed across all possible $2^K$ hash buckets. This significantly accelerates retrieval, achieving speedups of up to $3\times$ compared to the Locality-Sensitive Hashing (LSH). We empirically demonstrate that our system is scalable, highly effective, and efficient in identifying short audio queries ($\leq 2$s), particularly at high noise and reverberation levels.

## 1 Introduction

In the ever-expanding landscape of multimedia content, vector search has become increasingly crucial for efficiently retrieving similar items based on content representations, often represented as vectors in high-dimensional spaces. While extensive research has been devoted to content-based retrieval in the image domain (Luo et al., 2023), the domain of audio retrieval, particularly for the audio fingerprinting tasks, still needs to be explored. Audio fingerprinting generates a content-based compact summary of an audio signal, facilitating efficient storage and retrieval of the audio content. This technique finds applications across various scenarios, such as music recognition (Wang et al., 2003), duplicate detection (Burges et al., 2005), copyright enforcement (Saadatpanah et al., 2020) and second-screen services (Lohmüller & Wolff, 2019).

The existing audio fingerprinting methods rely on efficient indexing algorithms (Jegou et al., 2010; Gong et al., 2012; Gionis et al., 1999). In particular, Locality-Sensitive Hashing (LSH) (Gionis et al., 1999) implicitly divides the space into lattices, such that similar points are grouped into the same hash bucket. During retrieval, LSH evaluates the similarity between the query and points in the same hash bucket. However, when applied to real-world data characterized by non-uniform distributions, LSH faces challenges. This includes the issue of unbalanced hashing (Gao et al., 2014), where some buckets remain empty while others become overfilled, leading to reduced retrieval accuracy and unexpected delays. Furthermore, LSH, being an unsupervised method, requires multiple bucket probes and hash table constructions to achieve satisfactory performance.

In contrast to conventional approaches that consider representation learning and indexing separate processes, Singh et al. (2023) introduced an approach combining both aspects. Their approach simultaneously learns robust representations and balanced hash codes. This approach achieves balanced hash buckets by solving a balanced clustering objective, utilizing the optimal transport (OT) formulation (Villani et al., 2009). However, when dealing with a large number of cluster centroids, this requires a very large transportation matrix. Consequently, the Sinkhorn-Knopp (SK) algorithm (Cuturi, 2013), used to solve regularized transport problems, exhibits slow convergence and introduces substantial overhead during training. Also, this approach defines cluster centroids as normal-

ized binary vectors on a hypersphere. However, these non-orthogonal centroids lead to overlapping or closely situated clusters, resulting in hash buckets lacking distinct separation. Consequently, it causes data points near bucket boundaries and their perturbations to be assigned to different hash buckets, thereby compromising retrieval performance.

While numerous prior studies (Zheng et al., 2020; Yang et al., 2017; Hoe et al., 2021) have explored the generation of balanced hash codes in the context of image retrieval, none of these algorithms addresses the utilization of all possible hash buckets for a given bit length while maintaining a balanced distribution across hash buckets.

Similar to Singh et al. (2023), we adopt a joint representation learning and indexing approach in this paper. In contrast to Singh et al. (2023), we use a novel method for learning balanced hash codes, namely normalizing flow (NF). NF is a generative model that transforms complex distributions into tractable ones through a series of invertible and differentiable mappings. Our approach employs the RealNVP (Dinh et al., 2016) normalizing flow model to transform pre-trained audio representations within an $\mathbb{R}^K$ space to assign each dimension to a bimodal Gaussian mixture distribution. The result is an overall distribution comprising $2^K$ balanced modes, where each mode effectively corresponds to a representative hash bucket. This methodology aligns with the balanced clustering objective, with cluster centroids being vertices of a $K$-dimensional hypercube to ensure distinct cluster separation. Furthermore, we introduce a cross-entropy-based loss as a regularization term, enhancing the likelihood of data points and their perturbations being assigned to the same hash bucket. Our model, named **FlowHash**, presents two noteworthy advantages: firstly, it provides a scalable and optimal solution to achieve balanced hash buckets as often required in hash-based indexing. Secondly, it facilitates efficient database indexing using a classical hash table, which diverges from the prevalent method of constructing multiple hash tables in LSH. As a result, our approach enhances both the effectiveness and efficiency of the retrieval process. Overall, the main contributions of our work are as follows:

- We introduce a novel application of NF in hash-based indexing, leveraging it to obtain scalable balanced hash codes that ensure uniform distribution across all possible hash buckets for a given bit length.

- We introduce a regularization loss term to enhance the robustness of the hash codes.

- We present an audio fingerprinting method to effectively and efficiently identify short audio snippets ($\leq$2s) in high noise and reverberant environments.

- Our emperical study shows the robustness and scalability of our approach. Furthermore, we demonstrate its superior performance compared to LSH and the recently introduced OT-based method by Singh et al. (2023).

## 2 RELATED WORK

**Audio fingerprinting**. The common approaches for audio fingerprinting transform audio segments into low-dimensional vectors, commonly referred to as *representations* or *fingerprints*. The existing methods can be categorized based on two key characteristics. Firstly, the approach employed to generate fingerprints can be either knowledge-based (Haitsma & Kalker, 2002; Wang et al., 2003; Ke et al., 2005) or machine-learning-based (Gfeller et al., 2017; Báez-Suárez et al., 2020). Secondly, the generated fingerprints can be either hash representations (Baluja & Covell, 2008; Wu & Wang, 2022) or real-value representations (Báez-Suárez et al., 2020; Singh et al., 2022). The real-value representations offer a notable advantage in precise identification owing to their expansive information-capturing capability. However, hash-based representations facilitate fast comparisons and impose lower memory demands. Recent advances in audio fingerprinting have witnessed the emergence of deep-learning-based methodologies (Chang et al., 2021; Singh et al., 2022; 2023), often trained within a self-supervised learning framework. These methods exhibit robustness and achieve high retrieval performance even under high-distortion conditions such as noise and reverberation.

**Hash-based indexing**. A large body of work exists on hash-based indexes for approximate nearest-neighbor searches in high-dimensional space. Locality-sensitive hashing methods, like Multi-Probe LSH (Lv et al., 2007) and Cross-polytope LSH (Andoni et al., 2015), utilize hash functions to map data to fixed-size hash codes, yielding sub-linear time complexity. The LSB-tree (Tao et al.,

2009) merges LSH with trees for logarithmic query complexity. Entropy-LSH (Panigrahy, 2005) improves similarity-preserving hashing through entropy-based techniques. Bayesian LSH (Satuluri & Parthasarathy, 2011) enhances search precision via probabilistic models. Deep learning-based LSH approaches, such as Deep-LSH (Gao et al., 2014) and Neural-LSH (Dong et al., 2019), leverage deep learning for effective hash function learning.

**Generative modeling**. The two prominent generative models include Generative Adversarial Networks (GANs) (Goodfellow et al., 2014) and Variational Autoencoders (VAEs) (Kingma & Welling, 2013), facilitating data generation and latent space exploration. However, neither of them evaluates explicit probability estimation on generated samples. Normalizing Flows (NFs) have emerged as a promising alternative. They employ invertible transformations to model complex distributions, yielding both tractable likelihood estimation and efficient sampling. NFs find several applications, including density estimation (Dinh et al., 2016), audio generation (Esling et al., 2019), and anomaly detection (Gudovskiy et al., 2022).

**Optimal transport**. OT has gained traction (Torres et al., 2021) in machine learning for its ability to quantify the minimal cost of transforming one distribution to another. It finds applications in diverse domains, such as generative modeling (Arjovsky et al., 2017), domain adaptation (Courty et al., 2017), and clustering (Caron et al., 2020). In machine learning, the Sinkhorn-Knopp algorithm (Cuturi, 2013) solves optimal transport problems efficiently. However, its iterative nature and matrix computations lead to computational complexity, which poses challenges for integration into machine-learning tasks, particularly in large-scale scenarios. Moreover, high memory requirements due to operations on large cost matrices further limit its practicality.

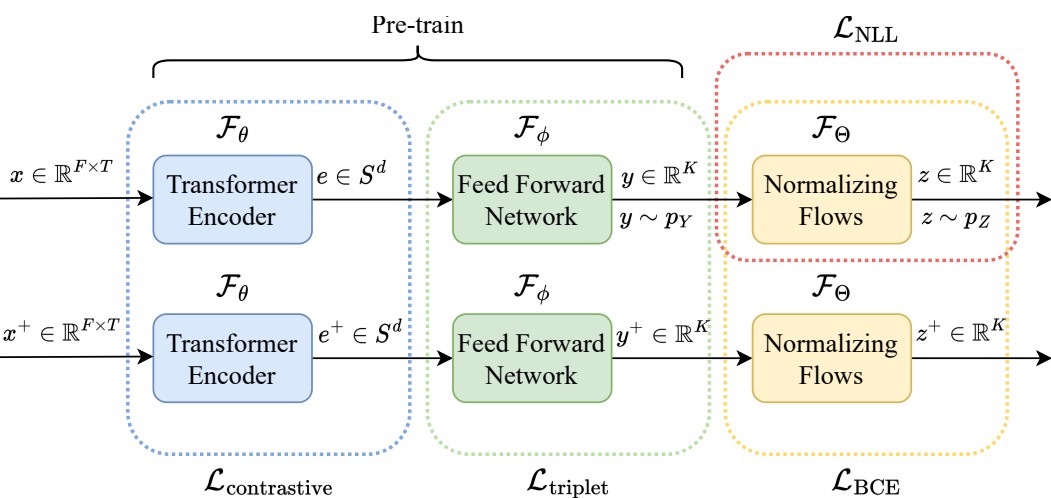

Figure 1: An overview of the FlowHash model. The encoders $\mathcal{F}_\theta$ and $\mathcal{F}_\phi$ are initially pre-trained in a self-supervised learning framework to generate robust encodings $e \in S^d$ and $y \in \mathbb{R}^K$, respectively, corresponding to the audio segment $x$. Subsequently, $y$, combined with a fixed distribution $p_Y$, is fed into the normalizing flow model $\mathcal{F}_\Theta$ to compute $z \in \mathbb{R}^K$. The distribution of $z$ conforms to the desired $p_Z$, facilitating the generation of balanced hash codes. Additionally, we introduce a regularization term during normalizing flow training to enhance code robustness.

## 3 OUR APPROACH: FLOWHASH

Our primary goal is the efficient indexing of the fingerprint database of size $N$, denoted as $E = \{e_n\}_{n=1}^N$, using a balanced hash table $T = \{h_k : A_k, \ A_k \subset E, \ k \in \{1, 2, ..., 2^K\}\}$. To this end, we first project fingerprints $e$ into a low-dimensional $\mathbb{R}^K$ space while preserving the neighborhood structure. These projections are subsequently transformed using the normalizing flow model to achieve a distribution characterized by $2^K$ well-balanced modes, each linked to a distinct hash code. This ensures that the size of each $A_k$ is approximately $N/2^K$. We illustrate an overview of our method in Figure 1.

### 3.1 ENCODER: REPRESENTATION LEARNING

We utilize the Transformer-based encoder $\mathcal{F}_\theta$ as initially proposed by Singh et al. (2023) to obtain contextualized audio representations. Given a set $\mathcal{D}$ of audio files, we randomly select an audio segment $x$ of fixed length and convert it to a log-Mel spectrogram. We further split the spectrogram into non-overlapping patches along the temporal axis and convert each into a 1D embedding using a projection layer. The sequence of these embedding are then fed into the Transformer encoder. The Transformer output sequence is then concatenated, thereby preserving the temporal structure of the audio. This concatenated sequence is then projected to a $d$-dimensional embedding, followed by length-normalization. This process results in a *fingerprint* denoted as $e = \mathcal{F}_\theta(x) \in S^d$. Similarly, we generate $e^+ = \mathcal{F}_\theta(x^+)$ for the distorted counterpart $x^+$ of $x$. To achieve robust representations, we train the encoder to maximize the cosine similarity between $e$ and $e^+$ using a contrastive loss:

$$\mathcal{L}_{\text{contrastive}} = -\log \frac{\exp\left((e \cdot e^+)/\tau\right)}{\exp\left((e \cdot e^+)/\tau\right) + \sum_{e^-} \exp\left((e \cdot e^-)/\tau\right)}, \tag{1}$$

where $\tau$ is a temperature hyperparameter that facilitates effective learning from hard negatives.

Next, we focus on learning the $K$-bit hash code $h$ corresponding to each fingerprint $e$. As a first step, we employ a projection mapping $\mathcal{F}_\phi$, which transforms the fingerprint into a low-dimensional $\mathbb{R}^K$ space, resulting in $y = \mathcal{F}_\phi(e)$. We learn this mapping using the triplet loss to ensure the respective projections of $e$ and $e^+$ maintain proximity based on the Euclidean distance as:

$$\mathcal{L}_{\text{triplet}} = \max(0, \| y - y^+ \|_2 - \| y - \frac{1}{2M-2} \sum_{m=1}^{2M-2} y_m^- \|_2) \tag{2}$$

Note that in a minibatch comprising $M$ pairs $\{x, x^+\}$, resulting in $2M$ samples, there are $2M$-2 negative samples, $x^-$, for each pair within the batch. Finally, we pre-train the encoders $\mathcal{F}_\theta$ and $\mathcal{F}_\phi$ using the overall loss that combines both contrastive and triplet losses:

$$\mathcal{L}_{\text{pretrain}} = \mathcal{L}_{\text{contrastive}} + \mathcal{L}_{\text{triplet}} \tag{3}$$

The reason for pre-training the encoders is to establish a fixed distribution $p_Y$ of projections $y$. This distribution serves as input to the normalizing flow model in the subsequent step.

### 3.2 NORMALIZING FLOW: BALANCED HASHING

We aim to transform samples $y$ into $z$ while ensuring that each component $z_k$ independently follows a bimodal Gaussian mixture distribution $p_{Z_k}$ along its respective $k$-th dimension. This transformation results in a joint distribution $p_Z$ consisting of $2^K$ balanced modes in the $\mathbb{R}^K$ space (see Figure 5 in Appendix A.4). To accomplish this, we leverage the RNVP normalizing flow model (Dinh et al., 2016).

The RNVP model applies a series of invertible mappings to transform a simple prior distribution into a complex distribution. Let $Z \in \mathbb{R}^K$ be a random variable with prior distribution $p_Z$, and $f_\Theta = f_L \circ f_{L-1} \circ ... \circ f_1$ be an invertible function such that $Y = f_\Theta(Z)$. We can determine the probability density of random variable $Y$ using the change of variable of formula as:

$$\log(p_Y(y)) = \log(p_Z(f_\Theta^{-1}(y))) + \sum_{l=1}^{L} \log |\det(J_{f_l^{-1}}(y_l))|, \tag{4}$$

where $J_{f_l^{-1}}$ is the Jacobian of $f_l^{-1}$. We denote the $l$-th intermediate flow output as $y_l = f_l \circ f_{l-1} \circ ... \circ f_1(z)$ and thus $y_L = y$. These mappings $f_l$ are parameterized by neural networks, and when applied sequentially, they are able to transform a simple distribution into a more complex one. Moreover, these mappings are designed to be invertible with tractable Jacobian determinants. In RNVP, we define these mappings as affine coupling layers. This layer splits the input $z \in \mathbb{R}^K$ into two disjoint parts, $(z^{(1)}, z^{(2)}) \in \mathbb{R}^k \times \mathbb{R}^{K-k}$ and transforms the input non-linearly as:

$$y^{(1)} = z^{(1)} \tag{5}$$

$$y^{(2)} = z^{(2)} \odot \exp(s(z^{(1)})) + t(z^{(1)}), \tag{6}$$

where $s(.)$ and $t(.)$ are neural networks, and $\odot$ represents the Hadamard product.

Here, we consider $z_k$ of $z$ as mutually independent components, each following a fixed bimodal Gaussian distribution $p_{Z_k}$ and thus we define $p_Z$ as:

$$\log(p_Z(z)) = \sum_{k=1}^{K} \log(p_{Z_k}(z_k)) = \sum_{k=1}^{K} \log(0.5 \cdot \mathcal{N}(z_k| - 2, 1) + 0.5 \cdot \mathcal{N}(z_k|2, 1)), \quad (7)$$

where $\mathcal{N}(z; \mu, \sigma^2)$ represents a Gaussian distribution with mean $\mu$ and variance $\sigma^2$. We train the RNVP model using the forward KL-divergence loss function, which is equivalent to minimizing the negative log-likelihood of the input data:

$$\mathcal{L}_{\text{NLL}} = -\log(p_Y(y)) \quad (8)$$

Here, in contrast to the generative task, we employ the inverse function $f_\Theta^{-1}$ to transform the input embedding $y$ into $z$. Subsequently, after training the model, we use it to generate a $K$-bit hash code $h$ by quantizing $z$ as $h = \text{positive}(z)$. Specifically, for each component $k = 1, 2, ..., K$, we define $\text{positive}(z_k) = 1$, if $z_k > 0$, and $\text{positive}(z_k) = 0$ otherwise.

**Hash codes robustness.** We also aim to ensure that both fingerprints in any given pair $\{e, e^+\}$ are consistently mapped to the same hash bucket $h$. This objective essentially translates to a binary classification problem, where each bit is assigned to either class 0 or 1. Therefore, we introduce cross-entropy based regularization term to achieve robust hash codes. The primary purpose of the regularization is to enforce the embeddings $z$ and $z^+$ to align closely, thereby promoting the reliable mapping of samples to hash buckets. The regularization term is defined as:

$$\mathcal{L}_{\text{BCE}} = \sum_{k=1}^{K} \left( \mathcal{H}(z_k, z_k^+) + \mathcal{H}(z_k^+, z_k) \right), \quad (9)$$

where $\mathcal{H}(u, v)$ represents the cross-entropy between two distribution $u$ and $v$. In our context, it quantifies the dissimilarity between $z_k$ and $z_k^+$. To further clarify, $\mathcal{H}(u, v)$ is computed as:

$$\mathcal{H}(u, v) = -\left[ p_{Z_{1k}}(u) \cdot \log(p_{Z_{1k}}(v)) + p_{Z_{2k}}(u) \cdot \log(p_{Z_{2k}}(v)) \right], \quad (10)$$

Here, $p_{Z_{1k}}$ and $p_{Z_{2k}}$ denote the probability density associated with two modes of the Gaussian mixture distribution along the $k$-th dimension. We finally train our model to minimize the overall loss combining negative log-likelihood loss and regularization term:

$$\mathcal{L}_{\text{NF}} = \frac{1}{M} \sum_{m=1}^{M} \left( \mathcal{L}_{\text{NLL}}(z_m) + \lambda \mathcal{L}_{\text{BCE}}(z_m, z_m^+) \right), \quad (11)$$

where the hyperparameter $\lambda \geq 0$ controls the trade-off between balanced hashing and hash code robustness.

### 3.3 INDEXING

We employ the trained encoders and normalizing flow model to extract fingerprints from a set of reference audio tracks $\mathcal{D}$ and subsequently index them using a hash table $T$ for an efficient retrieval during inference:

- **Fingerprinting**: We first extract audio segments of $t$ seconds from each audio track at a regular interval of $s(< t)$ seconds. Then, we generate a log-Mel spectrogram corresponding to each segment and feed it into the encoder to generate its fingerprint $e \in S^d$.

- **Hashing**: We input the fingerprint $e$ into the projection encoder, followed by the normalizing flow model, which computes $z \in \mathbb{R}^K$. To create the hash code $h \in \{0, 1\}^K$, we quantize $z$ using the positive$(.)$ function. However, this straightforward approach has a limitation - it assigns a bit with full confidence even to a $z$ value near 0. Moreover, their slightly perturbed counterpart $z^+$ may map to a different bit than that assigned to $z$, leading to a degradation in retrieval performance. Therefore, we assign both bit 0 and 1 to $z$ values within a small neighborhood $r(> 0)$ around 0. As a result, $e$ is mapped to multiple hash codes $h(z) = \{h_1, h_2, \ldots, h_i\}$. This ensures a more robust retrieval performance, mitigating the potential performance loss associated with near-zero $z$ values.

- **Hash Table**: Finally, we build the hash table $T = \{h_k : \{e_n \mid h_k \in h(z_n)\}, n = 1, 2, \ldots, N\}$, where $N$ is the total number of fingerprints extracted from all reference audio tracks.

### 3.4 Retrieval

For a query $q$ of length $\tilde{t}$ $(\geq t)$ seconds, we follow the same steps as above to extract fingerprints $e_q = \{e_{q_o}\}_{o=1}^{O}$, where $O$ is the number of segments extracted from $q$. For each $e_{q_o}$, we compute its corresponding hash codes $h(q_o) = \{h_1, \ldots, h_i\}$ depending on a neighborhood $\tilde{r}$. Then, we look up into each hash bucket to retrieve all the bucket candidates, $C = \{T(h_k), h_k \in h(q_o)\}$. To find the best match among the candidates, we perform a linear search as: $\arg\max_{p \in C} (e_p \cdot e_{q_o})$

As the query is segmented sequentially over time, the retrieved set of best-matching indices must correspond to a contiguous sequence. However, this contiguity is not always achieved due to approximate matching. To this end, we employ a simple yet effective subsequence search strategy (see Figure 6 in Appendix). This approach enables us to precisely locate the best-matching subsequence within the reference database that aligns with the query $q$. Let $i = \{i_1, \ldots, i_O\}$ be the retrieved sequence of indices. Then, we generate all possible sequence candidates as:

$$S_m = \{i_o + o - m, o = 1, 2, .., O\}, m = 1, 2, \ldots, O \tag{12}$$

Finally, we select the best matching sequence $S_{m^*}$ which has the maximum consensus with $i$:

$$m^* = \arg\max_m \sum_{o=1}^{O} I_{[S_{m,o}=i_o]} \tag{13}$$

## 4 Experiments

### 4.1 Dataset

**Music**: We use the Free Music Archive (FMA) (Defferrard et al., 2016) as a benchmark dataset prominently used for audio fingerprinting tasks. The dataset comprises three primary subsets: *small*, *medium*, and *large*, each containing 30s audio clips. We use the *small* subset for model training, which consists of 8000 balanced clips representing the top 8 genres in the dataset. We use the *medium* set of 25,000 clips, equivalent to 208 hours, to evaluate systems performance. We also use the *large* subset comprising a substantial 106,574 clips, totaling about 888 hours of content and spanning 161 genres, to assess the scalability of the systems. Note that the *medium* and *large* subsets exhibit an unbalanced track distribution per genre. This diversity in genre representation allows for a more comprehensive assessment of system performance on a larger scale.

**Noises**: We extracted diverse noise samples from the MUSAN corpus (Snyder et al., 2015) for model training. We used a distinct set of noise clips obtained from the ETSI database [1] during the evaluation phase. These clips correspond to various environmental contexts, including babble, cafeteria, car, living room, shopping, train station, and traffic.

**RIRs**: We used the MUSAN corpus to acquire Room Impulse Responses (RIRs) corresponding to various environmental settings, ranging from small indoor rooms to large rooms such as halls, conference rooms, and churches. We used RIRs from the Aachen Impulse Response Database (Jeub et al., 2009) during the evaluation. These RIRs correspond to a $t_{60}$ reverberation time of 0.2 to 0.8s.

### 4.2 Metrics

**Efficacy** is measured using the top-1 recall rate, which indicates the percentage of outcomes where the correct match is found at the top rank. Note that we consider the correct match only if the identified timestamp of a query is within 50ms of the actual timestamp.

**Efficiency** serves as a quantitative measure of how effectively the database is searched to find the top match. Let $N$ denote the total number of database samples and $N_p$ be the total unique points evaluated in the candidates list $C$, then we define the metric as $\text{eval} = 100 \times \frac{N_p}{N}$.

### 4.3 Baselines

We compare our proposed method, FlowHash, against state-of-the-art deep learning-based encoding methods introduced by Singh et al. (2022) (AE) and Chang et al. (2021) (NAFP). Moreover, we

---

[1] https://docbox.etsi.org/

employ the recently proposed OT-based method (Singh et al., 2023) for balanced hashing and the LSH as baseline methods to show the effectiveness of our proposed approach for efficient retrieval.

## 4.4 IMPLEMENTATION

**Augmentation**: We randomly apply the following distortions to audio input $x$ to generate $x^+$ as:

- Noise: We add a randomly selected background noise within a 0-20dB SNR level range.
- Reverberation: We filter the input audio with a randomly chosen RIR to simulate room acoustics.
- Time offset: We add a temporal offset of up to 50ms to account for potential temporal inconsistencies in the real-world scenario.

**Database**: We use the *medium* and *large* subsets to construct their respective fingerprint databases. We generate fingerprints for 1s audio segments extracted every 100ms in each audio track. As a result, the *medium* and large subsets yield a database of $\sim$7M and $\sim$29M fingerprints, respectively.

**Queries**: We generate 1,000 queries by randomly selecting segments from the reference audio tracks. These queries vary in length from 1-5s and are distorted with added noise, reverb, or a mix of both. To generate noisy reverberant queries, we first filter them using an RIR with a $t_{60}$ of 0.5s, followed by adding noise in a 0-20dB SNR range.

## 4.5 RESULTS

Table 1: Comparison of top-1 recall rates (%) in various distortion environments for varied query lengths. The final column represent retrieval efficiency, determined by the percentage of the total database evaluated to identify the top match. We underline our results if the accuracy drop is less than 1.5% compared to the best performing baseline.

| | Method | Noise ↑ | | | | | Noise + Reverb ↑ | | | | | Reverb ↑ | | | | | eval ↓ |
|---|---|---|---|---|---|---|---|---|---|---|---|---|---|---|---|---|---|
| | | 0dB | 5dB | 10dB | 15dB | 20dB | 0dB | 5dB | 10dB | 15dB | 20dB | 0.2s | 0.4s | 0.5s | 0.7s | 0.8s | |
| 1s | NAFP + LSH | 50.1 | 66.4 | 73.0 | 75.1 | 76.0 | 21.3 | 43.1 | 53.9 | 58.3 | 60.5 | 61.4 | 60.3 | 57.6 | 48.5 | 42.3 | 2.28 |
| | AE + LSH | 59.1 | 70.2 | 71.5 | 74.8 | 75.7 | 32.1 | 52.0 | 57.9 | 63.3 | 65.1 | 66.2 | 65.3 | 63.1 | 55.3 | 51.9 | 2.32 |
| | TE + LSH | 74.6 | 81.9 | 87.0 | 89.1 | 91.4 | 47.8 | 67.5 | 77.8 | 82.1 | 83.0 | 83.4 | 82.2 | 80.2 | 74.0 | 72.7 | 2.30 |
| | TE + OT | 65.3 | 82.0 | 87.5 | 89.6 | 90.2 | 42.3 | 62.1 | 74.1 | 78.7 | 81.3 | 83.5 | 82.7 | 80.1 | 68.0 | 63.4 | 1.21 |
| | TE + NF (FlowHash) | 63.2 | 81.0 | 89.1 | 93.0 | 92.7 | 36.6 | 60.1 | 73.2 | 78.9 | 81.4 | 82.5 | 79.5 | 79.2 | 70.6 | 68.5 | 0.76 |
| 2s | NAFP + LSH | 69.7 | 79.0 | 85.0 | 86.8 | 87.1 | 40.1 | 63.3 | 72.6 | 75.0 | 76.1 | 77.1 | 76.7 | 75.5 | 68.3 | 61.0 | 2.28 |
| | AE + LSH | 75.4 | 83.7 | 84.5 | 87.0 | 87.7 | 53.1 | 71.6 | 78.0 | 81.6 | 82.2 | 84.0 | 82.5 | 81.6 | 75.2 | 70.8 | 2.32 |
| | TE + LSH | 82.6 | 90.0 | 92.3 | 93.8 | 94.6 | 61.5 | 81.2 | 85.5 | 86.3 | 87.1 | 90.0 | 88.6 | 87.0 | 80.6 | 76.9 | 2.30 |
| | TE + OT | 81.6 | 90.0 | 92.6 | 93.6 | 94.2 | 59.2 | 80.6 | 85.1 | 86.1 | 87.3 | 91.6 | 88.5 | 86.0 | 79.1 | 71.1 | 1.21 |
| | TE + NF (FlowHash) | 82.0 | 90.1 | 94.1 | 94.8 | 95.9 | 55.9 | 78.6 | 84.7 | 85.9 | 86.9 | 90.2 | 87.6 | 87.6 | 79.6 | 74.9 | 0.76 |
| 3s | NAFP + LSH | 77.0 | 83.8 | 88.0 | 88.7 | 89.1 | 53.6 | 71.1 | 77.7 | 78.5 | 81.0 | 82.3 | 78.6 | 76.5 | 69.1 | 62.3 | 2.28 |
| | AE + LSH | 81.7 | 86.5 | 88.1 | 88.7 | 89.4 | 64.4 | 78.1 | 82.4 | 85.0 | 86.7 | 88.8 | 85.6 | 84.0 | 76.7 | 72.3 | 2.32 |
| | TE + LSH | 86.5 | 92.2 | 94.1 | 94.9 | 96.0 | 70.6 | 85.5 | 88.7 | 88.6 | 89.3 | 92.8 | 89.6 | 89.0 | 81.3 | 78.2 | 2.30 |
| | TE + OT | 85.0 | 91.1 | 95.0 | 96.0 | 96.6 | 70.0 | 83.8 | 88.4 | 88.7 | 89.4 | 93.1 | 91.0 | 87.2 | 79.5 | 73.1 | 1.21 |
| | TE + NF (FlowHash) | 86.0 | 92.1 | 94.7 | 96.0 | 96.7 | 69.6 | 84.6 | 88.9 | 88.7 | 89.7 | 92.3 | 88.9 | 88.0 | 80.1 | 77.0 | 0.76 |
| 5s | NAFP + LSH | 81.4 | 88.2 | 90.3 | 91.5 | 91.6 | 62.1 | 80.1 | 82.5 | 82.7 | 83.6 | 84.5 | 83.1 | 78.1 | 74.0 | 64.8 | 2.28 |
| | AE + LSH | 83.1 | 89.0 | 90.8 | 91.2 | 92.0 | 76.3 | 85.1 | 87.1 | 89.0 | 89.1 | 90.5 | 88.0 | 86.3 | 80.0 | 76.9 | 2.32 |
| | TE + LSH | 89.9 | 92.4 | 95.4 | 96.4 | 96.5 | 81.3 | 90.0 | 90.2 | 90.9 | 89.7 | 94.0 | 91.2 | 90.1 | 82.3 | 79.0 | 2.30 |
| | TE + OT | 88.1 | 93.4 | 95.3 | 96.1 | 97.0 | 80.1 | 88.2 | 90.8 | 90.8 | 91.0 | 95.2 | 91.4 | 88.3 | 79.4 | 74.9 | 1.21 |
| | TE + NF (FlowHash) | 89.9 | 93.2 | 96.0 | 96.4 | 97.0 | 80.6 | 90.0 | 90.8 | 90.9 | 91.2 | 94.2 | 90.3 | 90.2 | 81.5 | 78.9 | 0.76 |

Table 1 compares our method with the baselines, focusing on efficiency and efficacy. We first assess the effectiveness by comparing the performance of the Transformer-based encoding (TE) with NAFP and AE methods while using LSH for indexing and maintaining a consistent evaluation of points. TE excels at capturing contextual information, resulting in better discriminative embeddings. Thus, TE consistently achieves 10-20% higher hit rates across varying distortion levels and query lengths.

For 1 second queries, we observed several mismatches due to repeated instances of a query (e.g. refrain of a song) in an audio track. Therefore, it becomes crucial to choose an extended query to be more discriminative to accurately identify its correct match. This is corroborated by the consistent trend of improved performance across all methods with extended query lengths. Moreover, our subsequence search allows precise query alignment in the identified audio track, resulting in a substantial 10-20% increase in recall rates for longer queries.

Furthermore, we compare the retrieval efficiency of our proposed approach using NF with OT and LSH on the fingerprints database generated with TE. Our method outperforms LSH and OT across different distortion levels, achieving an average speedup of 3.1× and 1.6×, respectively. With this

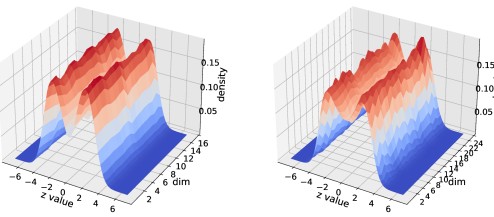

Figure 2: The illustration shows the marginal distributions, $p_{Z_k}$, for each dimension. Each dimension exhibits a bimodal Gaussian mixture distribution for $K$=16 (left) and $K$=24 (right). This indicates that the joint distribution $p_Z$ consists of well-balanced $2^K$ modes in the $\mathbb{R}^K$ space.

speedup, we achieve competitive or even higher hit rates across different distortion environments. However, our method underperforms in noisy reverberant environment, particularly at 0dB and 5dB SNR for the short (1 second) queries. Nevertheless, it is encouraging that the accuracy gap narrows to 1-2% for longer query lengths and even surpasses the accuracy of the other methods at high SNR levels.

It is important to highlight that OT selects top-$k$ buckets to probe by evaluating the similarity of a query with all $2^K$ hash buckets, which adds significant computational overhead in the retrieval process as $K$ increases. In contrast, our method efficiently probes buckets based on near zero-valued $z$-values, resulting in a computationally inexpensive procedure. On average, our method probes 35 buckets, which is a substantial reduction compared to 1000 bucket probes in OT and LSH.

## 4.6 ANALYSES

**Target distribution**. In Figure 2, we present an analysis for $K$=16, demonstrating that the normalizing flow effectively yields the target distribution $p_Z$. In particular, we analyze the marginal distributions $p_{Z_k}$ of $p_Z$ along each dimension. These marginals are depicted as a bimodal Gaussian mixture with balanced modes, resulting in an overall balance across all $2^{16}$ modes in the $\mathbb{R}^{16}$ space. Notably, we achieve this balance mode distribution even for a larger $K$, such as $K$=24. This highlights the applicability of our approach in scenarios where a substantial number of hash buckets are required to index an extensive database efficiently.

**Balanced hash codes**. We evaluate the balance of $K$-bit hash codes by examining the density of each hash bucket within a hash table $T$. The density measures the proportion of total samples $N$ mapped to a hash bucket $h_k$, and is defined as $\rho_k = -\log_2(|T(h_k)|/N)$. A uniform distribution of density values indicates an optimal balance, each attaining the value of $K$. This signifies an ideal scenario where samples are evenly distributed across all possible $2^K$ hash buckets. Deviations in density from $K$, either higher or lower, indicate that the respective bucket is either underfilled or overfilled. We show in Figure 3 that NF achieves hash codes with a more balanced distribution compared to the OT formulation. Our method results in ~95% of total hash buckets that are almost uniformly filled, as opposed to ~78% in the OT-based approach. Adding the $\mathcal{L}_{\text{BCE}}$ loss introduces some disruption to this balance; however, it still outperforms the OT method.

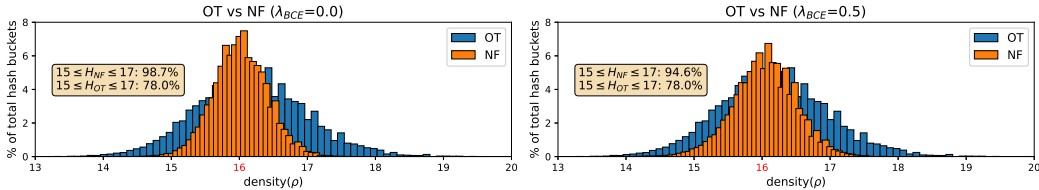

Figure 3: Comparison of hash code balance between NF and OT for $K$=16.

**Regularization loss**. To assess the effectiveness of the regularization term, we analyze the absolute value difference between $z_k$ and $z_k^+$ across all $K$ dimensions. Figure 4 shows that $z_k$ and $z_k^+$ tend to lie closer when employing the $\mathcal{L}_{\text{BCE}}$ loss function during training. Consequently, this increases the probability of $z_k$ and $z_k^+$ being assigned to the same bit and thus requires fewer bucket probes during the search. On the contrary, a substantial difference in $z$ values is observed without $\mathcal{L}_{\text{BCE}}$ loss. Our analysis indicates that ~82% of all pairs $(z, z^+)$ exhibit less than a 0.5 $z$-value difference, compared to only ~51% in the absence of $\mathcal{L}_{\text{BCE}}$.

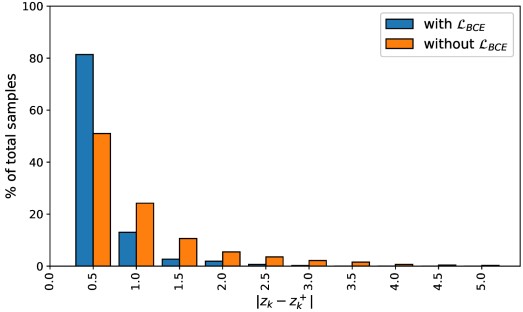

Figure 4: Effect of $\mathcal{L}_{\text{BCE}}$ on the absolute difference between $z_k$ and $z_k^+$. The addition of the $\mathcal{L}_{\text{BCE}}$ loss coaxes $z_k$ and $z_k^+$ to map them to the same bit, whereas its absence results in an increased distance between them.

**Scalability.** To evaluate the scalability of our method, we index a database comprising $\sim$29M fingerprints using 24-bit hash codes. Due to computational constraints, computing 24-bit hash codes with OT is infeasible. Therefore, we present the results for both LSH and our method in Table 2. Both methods experience a decline in recall rate accuracy, ranging from 10-20%, in noisy and noisy+reverberant environments, particularly at 0dB and 5dB SNR levels. However, at lower distortion levels, we observe a more modest decrease in accuracy, ranging from 5-7%. In comparison to LSH, our method attains a similar hit rate while exhibiting a 2.4$\times$ speedup, which is slightly lower than the 3.1$\times$ speedup observed in the prior evaluation conducted on a smaller database.

Table 2: Comparison of top-1 recall rates (%) and the percentage of database (large) evaluated in various distortion environments for 1 second queries. We underline our results if the accuracy drop is less than 1.5% compared to the best performing baseline.

|  | Method | Noise ↑ | | | | | Noise + Reverb ↑ | | | | | Reverb ↑ | | | | | eval ↓ |
|---|---|---|---|---|---|---|---|---|---|---|---|---|---|---|---|---|---|
|  |  | 0dB | 5dB | 10dB | 15dB | 20dB | 0dB | 5dB | 10dB | 15dB | 20dB | 0.2s | 0.4s | 0.5s | 0.7s | 0.8s |  |
| 1s | TE + LSH | 61.2 | **74.4** | **84.8** | 88.2 | 89.6 | **27.9** | **55.2** | **64.4** | 69.7 | 71.9 | **75.8** | 75.7 | **72.3** | **66.0** | **62.3** | 0.14 |
|  | TE + NF (FlowHash) | 51.5 | 73.5 | 84.3 | **89.9** | **90.1** | 23.5 | 48.4 | 63.5 | **71.0** | **75.4** | 75.1 | 75.6 | 71.2 | 63.6 | 59.5 | **0.06** |

## 5 LIMITATIONS

The main limitation of the proposed method lies in the computational complexity associated with generating balanced hash codes using NF. However, the balanced $K$-bit hash codes speed up the retrieval process, particularly as $K$ increases. Additionally, our method utilizes a computationally intensive encoder to generate fingerprints, enabling precise matching of queries even in high-distortion environments. While these choices do come with a trade-off in terms of longer encoding time, they ultimately contribute to the great efficacy and retrieval efficiency of our method. In terms of memory requirements, each fingerprint is encoded using 4 bytes, occupying a total of 4.8GB for $\sim$7M fingerprints. However, this space requirement could be reduced by half by encoding each fingerprint using 2 bytes with only a negligible ($\leq 0.5\%$) decline in the recall accuracy.

## 6 CONCLUSION

This paper proposes a novel application of NF in the domain of vector search, particularly for the audio fingerprinting task. We leverage normalizing flows to attain balanced $K$-bit hash codes. We achieve this by transforming vectors within a latent $\mathbb{R}^K$ space, resulting in a distribution characterized by well-balanced $2^K$ modes, each corresponding to a hash bucket. This allows an efficient database indexing using a balanced hash table. In addition, we incorporate a regularization term while training the NF model to ensure a vector and its corresponding perturbation map to the same hash bucket, thereby adding robustness to the indexing process. Moreover, we validate that our method produces hash codes with superior balancing compared to the recently proposed OT-based approach. We employ a self-supervised learning framework to enhance robustness of our fingerprinting system against high noise and reverberation levels. Furthermore, our system demonstrates scalability and efficiency in retrieval, surpassing the performance of both the LSH and the OT-based approach.

## 7 REPRODUCIBILITY

We provide the details of the models used to build our system in the Appendix A.3. Additionally, we intend to make the source codes available to the public for reproducibility after the review process.

### ACKNOWLEDGMENTS

Use unnumbered third level headings for the acknowledgments. All acknowledgments, including those to funding agencies, go at the end of the paper.

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

# A APPENDIX

## A.1 MORE RESULTS

### A.1.1 LIBRISPEECH

We also demonstrate in Table 3 the performance of our method on the LibriSpeech database, commonly used for the Automatic Speech Recognition (ASR) task. This database provides free audiobooks of the public domain text and consists of recordings from a diverse set of readers. This database is organized into distinct sets for training, development, and testing. We used the 'train-clean-100' training subset, consisting of 100 hours of data for training our models. We used the 'train-clean-360' subset for evaluation purposes, containing 360 hours of data. This subset is employed to build the fingerprints database, resulting in ∼12M fingerprints.

Table 3: Comparison of top-1 recall rates (%) in various distortion environments for 1 second queries. The final column represent retrieval efficiency, determined by the percentage of the total database evaluated to identify the top match. We underline our results if the accuracy drop is less than 1.5% compared to the best performing baseline.

|  | Method | Noise ↑ | | | | | Noise + Reverb ↑ | | | | | Reverb ↑ | | | | | eval ↓ |
|---|---|---|---|---|---|---|---|---|---|---|---|---|---|---|---|---|---|
|  |  | 0dB | 5dB | 10dB | 15dB | 20dB | 0dB | 5dB | 10dB | 15dB | 20dB | 0.2s | 0.4s | 0.5s | 0.7s | 0.8s |  |
|  | TE + LSH | **84.1** | **90.6** | 90.5 | 92.7 | 93.5 | **70.9** | 79.7 | 88.7 | 89.6 | 89.4 | 91.9 | 91.2 | 90.7 | 89.5 | **89.2** | 1.98 |
| 1s | TE + OT | 75.3 | 90.0 | 91.2 | 92.9 | 93.2 | 62.3 | **80.3** | 85.0 | 89.4 | 93.2 | 93.1 | 91.7 | **91.5** | **90.7** | 83.4 | 1.01 |
|  | TE + NF (FlowHash) | 75.5 | 90.2 | **92.6** | **93.5** | **94.4** | 57.6 | 80.1 | **86.8** | **91.7** | **93.4** | **93.5** | **91.9** | 91.2 | 90.2 | 87.5 | **0.58** |

## A.1.2 BROADCAST CONTENT

In Table 4, we compare the performance of our method using an internally curated dataset of broadcast content. This dataset encompasses diverse content types such as awards shows, folk music, sports programs, autobiographies, and interviews. To train our model, we utilized a subset of approximately 100 hours of data. The remaining dataset, totaling around 200 hours of content, was used to construct a fingerprint database for evaluation purposes. Our reference database comprised of ∼7M fingerprints.

Table 4: Comparison of top-1 recall rates (%) in various distortion environments for 1 second queries. The final column represent retrieval efficiency, determined by the percentage of the total database evaluated to identify the top match. We underline our results if the accuracy drop is less than 1.5% compared to the best performing baseline.

|  | Method | Noise ↑ | | | | | Noise + Reverb ↑ | | | | | Reverb ↑ | | | | | eval ↓ |
|---|---|---|---|---|---|---|---|---|---|---|---|---|---|---|---|---|---|
|  |  | 0dB | 5dB | 10dB | 15dB | 20dB | 0dB | 5dB | 10dB | 15dB | 20dB | 0.2s | 0.4s | 0.5s | 0.7s | 0.8s |  |
|  | TE + LSH | **76.1** | **85.4** | 89.2 | 90.7 | 91.6 | **54.5** | **73.7** | **82.2** | 82.9 | 84.2 | 90.0 | 87.7 | 85.4 | **83.5** | 77.2 | 2.01 |
| 1s | TE + OT | 70.0 | 85.0 | **90.0** | 91.7 | 92.9 | 49.8 | 70.0 | 82.0 | 86.4 | 88.0 | **90.2** | **88.5** | 84.5 | 81.9 | 78.4 | 1.06 |
|  | TE + NF (FlowHash) | 68.2 | 84.7 | 89.6 | **92.2** | **93.4** | 45.9 | 70.8 | 81.8 | **87.4** | **88.0** | 89.4 | 87.9 | **86.8** | 82.7 | **78.7** | **0.80** |

## A.2 ABLATION

### A.2.1 INDEXING

In the indexing phase (section 3.3), we proposed assigning fingerprint $e$ to multiple hash buckets depending on the width parameter $r$ ($r > 0$). Similarly, during the inference phase, we use the width parameter $\tilde{r}$ to determine the hash buckets to probe for a given query fingerprint $e_{q_o}$. In Table 5, we show how varying $r$ and $\tilde{r}$ affect the probability of the candidate set $C$ containing the best match for query $q$ during lookup. Additionally, we investigate how the fingerprint dimensions $d$ influence this probability.

We find that increasing values for $r$ and $\tilde{r}$ result in higher probabilities across different distortion levels. However, this also entails a higher number of candidate evaluations because increasing $r$ leads to an increase in hash bucket size, and increasing $\tilde{r}$ requires probing more buckets during search. Therefore, to strike a balance between accuracy and the total candidate evaluation, we opted for $r = 1.0$ and $\tilde{r} = 1.25$ during the evaluation. The results also indicate that increasing the fingerprint dimension has negligible effect on the probability.

Table 5: The effect of widths $r$ and $\tilde{r}$ and the fingerprints dimension $d$ at different distortion levels on the probability depicting that candidate list $C$ contains the best match for a query $q$.

| | Noise (dB) | | | | | | | | | | | | | | | | | | | |
|---|---|---|---|---|---|---|---|---|---|---|---|---|---|---|---|---|---|---|---|---|
| | r = 0.5, r̃ = 1.0 | | | | | r = 0.5, r̃ = 1.5 | | | | | r = 1.0, r̃ = 1.0 | | | | | r = 1.0, r̃ = 1.5 | | | | |
| | 0 | 5 | 10 | 15 | 20 | 0 | 5 | 10 | 15 | 20 | 0 | 5 | 10 | 15 | 20 | 0 | 5 | 10 | 15 | 20 |
| $d$=128 | 0.51 | 0.69 | 0.85 | 0.93 | 0.98 | 0.68 | 0.84 | 0.93 | 0.98 | 0.99 | 0.64 | 0.83 | 0.93 | 0.97 | 0.99 | 0.78 | 0.92 | 0.97 | 0.99 | 0.99 |
| $d$=256 | 0.50 | 0.71 | 0.85 | 0.94 | 0.98 | 0.70 | 0.96 | 0.93 | 0.98 | 0.99 | 0.65 | 0.84 | 0.93 | 0.98 | 0.99 | 0.81 | 0.92 | 0.97 | 0.98 | 0.99 |
| $d$=512 | 0.51 | 0.70 | 0.86 | 0.94 | 0.98 | 0.70 | 0.86 | 0.94 | 0.98 | 0.99 | 0.67 | 0.85 | 0.94 | 0.98 | 0.99 | 0.82 | 0.93 | 0.97 | 0.99 | 0.99 |
| | Noise + Reverb (dB) | | | | | | | | | | | | | | | | | | | |
| | r = 0.5, r̃ = 1.0 | | | | | r = 0.5, r̃ = 1.5 | | | | | r = 1.0, r̃ = 1.0 | | | | | r = 1.0, r̃ = 1.5 | | | | |
| | 0 | 5 | 10 | 15 | 20 | 0 | 5 | 10 | 15 | 20 | 0 | 5 | 10 | 15 | 20 | 0 | 5 | 10 | 15 | 20 |
| $d$=128 | 0.29 | 0.40 | 0.53 | 0.58 | 0.64 | 0.48 | 0.61 | 0.70 | 0.78 | 0.81 | 0.46 | 0.57 | 0.68 | 0.76 | 0.77 | 0.61 | 0.73 | 0.82 | 0.87 | 0.91 |
| $d$=256 | 0.30 | 0.42 | 0.52 | 0.61 | 0.65 | 0.50 | 0.62 | 0.74 | 0.79 | 0.83 | 0.47 | 0.60 | 0.70 | 0.76 | 0.79 | 0.61 | 0.75 | 0.83 | 0.89 | 0.91 |
| $d$=512 | 0.29 | 0.41 | 0.51 | 0.60 | 0.64 | 0.48 | 0.60 | 0.71 | 0.77 | 0.83 | 0.46 | 0.60 | 0.72 | 0.77 | 0.81 | 0.63 | 0.77 | 0.84 | 0.90 | 0.91 |

### A.2.2 ENCODING

We analyze the effect of increasing the fingerprint dimension, specifically from 64 to 128, 256, and 512. The results in Table 6 indicate that increasing dimensions from 64 to 128 leads to a 2-3% increase in the recall rate, particularly at low SNR levels in noisy and noisy reverberant environments. There is a negligible performance improvement when increasing dimensions from 128 to 256. However, a 3-5% gain in the recall rate is attained in both noisy reverberant and reverberant environments when dimensions are increased from 128 to 512, with negligible impact in the noisy environment. Note that the percentage of candidates evaluated is similar across different dimensions.

Table 6: Comparison of top-1 recall rates (%) in various distortion environments for varied fingerprints dimensions $d$.

| | Noise ↑ | | | | | Noise+Reverb ↑ | | | | | Reverb ↑ | | | | | eval ↓ |
|---|---|---|---|---|---|---|---|---|---|---|---|---|---|---|---|---|
| | 0dB | 5dB | 10dB | 15dB | 20dB | 0dB | 5dB | 10dB | 15dB | 20dB | 0.2s | 0.4s | 0.5s | 0.7s | 0.8s | |
| $d$=64 | 59.5 | 78.0 | 87.4 | 91.2 | 93.0 | 34.1 | 57.6 | 71.8 | 76.6 | 79.4 | 81.5 | 79.0 | 79.5 | 69.1 | 67.2 | 0.79 |
| $d$=128 | 63.2 | 80.3 | 89.1 | 93.0 | 92.7 | 36.6 | 60.1 | 73.2 | 78.9 | 81.4 | 82.5 | 79.5 | 79.2 | 70.6 | 68.5 | 0.76 |
| $d$=256 | 62.5 | 80.3 | 88.5 | 92.7 | 93.2 | 36.1 | 61.0 | 74.5 | 79.6 | 82.1 | 82.8 | 80.3 | 80.1 | 72.1 | 69.5 | 0.77 |
| $d$=512 | **64.9** | **82.9** | **89.0** | **92.6** | **93.4** | **41.0** | **64.5** | **77.8** | **82.9** | **84.8** | **86.2** | **82.3** | **82.1** | **73.7** | **71.3** | **0.77** |

### A.3 IMPLEMENTATION DETAILS

### A.3.1 MODEL ARCHITECTURE

- **Transformer Encoder:** We first divided the input spectrogram of 64 frequency bins and 100 time-frames into 10 non-overlapping patches along the temporal axis, each of dimension $64\times10$. These patches were transformed into embeddings of size 128 using a neural network. The sequence of these embeddings served as input to the Transformer model. The parameters of the Transformer encoder are as follows:
    - Number of layers: 8
    - Attention heads: 8
    - Feedforward neural network: 1 hidden layer with a size of 2048
    - Dimensionality: 128
    - Activation: ReLU
- **Projection Layer:** A feedforward network with a hidden layer of size 512 with ReLU activation
- **Normalizing Flow:** The following parameters were used to implement the RNVP model:
    - Number of coupling layers: 10
    - Neural network architecture in coupling layer: A feedforward network with 3 hidden layers of size 64 with ReLU activation

### A.3.2 OTHER DETAILS

- Sampling frequency (fs) of input audio: 16kHz
- Temperature $\tau$ in Equation 1: 0.1
- We chose $\lambda = 0.5$ to weight the $\mathcal{L}_{\text{BCE}}$ loss while training normalizing flow model.
- Batch size of $M$: 512 pairs

- We chose width $r = 1.0$ when indexing the database, whereas we chose $\tilde{r} = 1.25$ for the query during inference.
- We chose LSH implementation from `https://github.com/FALCONN-LIB/FALCONN`.

### A.3.3 TRAINING

The encoders were jointly trained using the Adam optimizer with a learning rate of 1e-4 for 1500 epochs. Further, we trained the RNVP model using the same optimizer with an initial learning rate of 1e-3 for 1200 epochs until the learning rate reached its minimum value of 1e-4. These models were trained on a single NVIDIA A100 GPU.

### A.4 NORMALIZING FLOW ON TOY DATASETS

We illustrate in Figure 5 the transformation of the 2D input data $Y$, representing different distributions, using the NF model. The NF transforms the data to follow a bimodal Gaussian distribution in each dimension, resulting in an overall distribution $p_Z$ consisting of 4 distinct modes. Each mode represents a cluster and thus can be encoded using a 2-bit hash code.

### A.5 FAILED APPROACHES

In the initial phase of the system development, we investigated two approaches to achieve balance clustering. Firstly, we looked into the Gumbel softmax trick (as used in Wav2Vec2.0) to learn a codebook. However, we observed a tendency for the codewords to collapse, rendering it unsuitable for our purposes. Subsequently, we investigated the Minimal-cost flow problem [2], a graph-based approach. While this approach enabled the attainment of balanced clusters, it proved to be computationally infeasible when employed, even when dealing with a relatively small number of clusters in the few hundred range. Moreover, we devised a custom loss function to achieve balanced clustering. However, none of the approaches yielded satisfactory results when a large number ($2^K$, where $K \geq 10$) of clusters were considered

---

[2]`https://www.ijcai.org/proceedings/2019/0414.pdf`

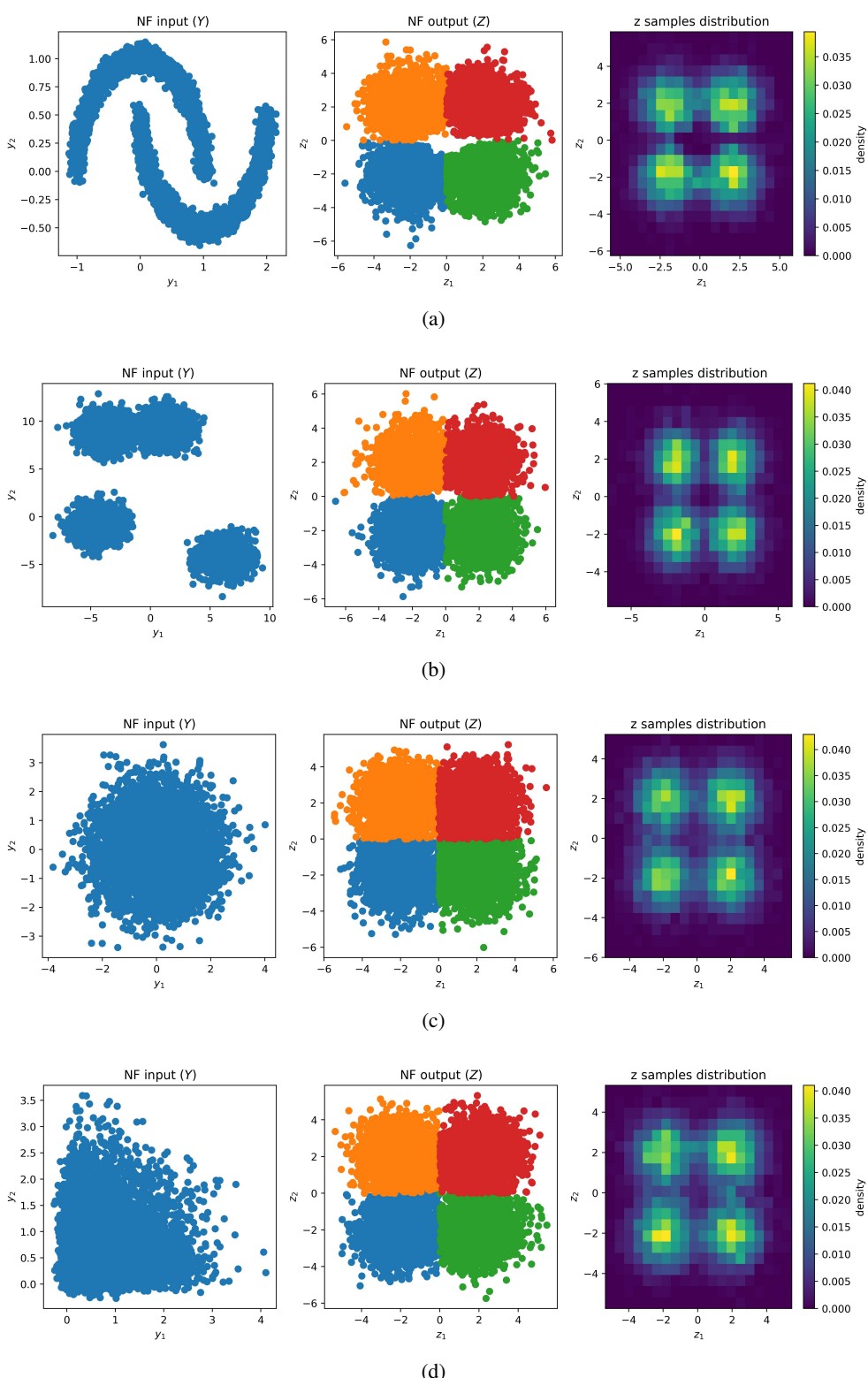

Figure 5: Examples illustrating the transformation of input 2D data (left) into the corresponding output (middle) that follows a bimodal Gaussian distribution along each dimension, resulting in an overall distribution consisting of 4 balanced modes (right).

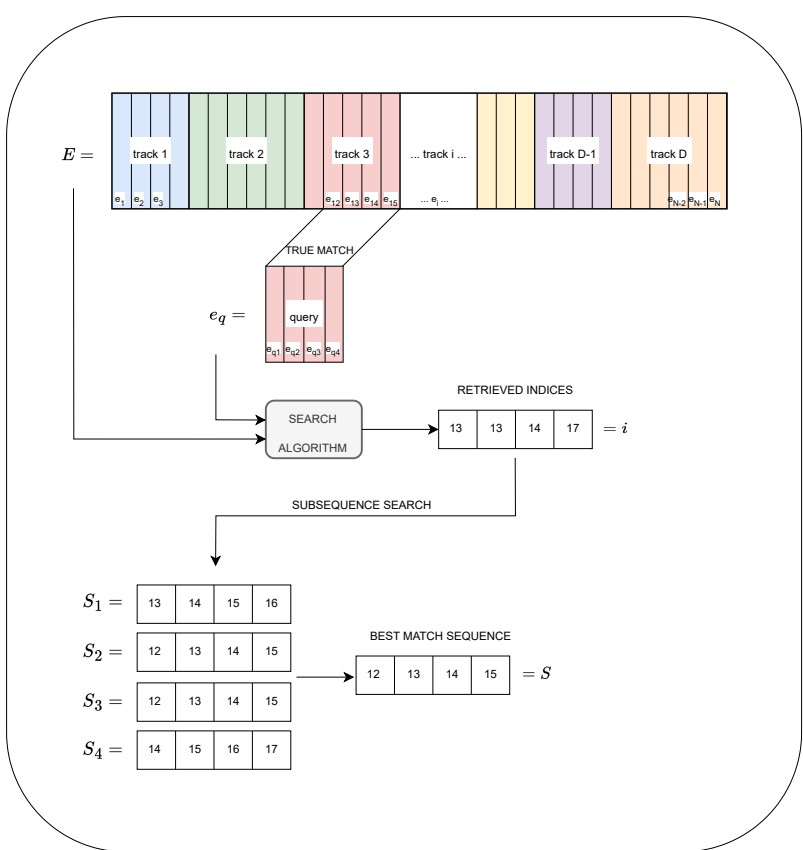

Figure 6: Illustration of the retrieval process using subsequence search.

