# OpenReview forum: "FlowHash: Accelerating Audio Search with Balanced Hashing via Normalizing Flow"
_ICLR.cc/2024/Conference — Submitted to ICLR 2024_

### Official Review · Reviewer_q2gt · 2023-10-24

**Soundness:** 2 fair
**Presentation:** 2 fair
**Contribution:** 1 poor
**Rating:** 3
**Confidence:** 4

**Summary:**

This paper proposes FlowHash, a framework for learning locality sensitive hashing codes for audio retrieval. It uses contrastive learning to distinguish between similar and dissimilar item pairs and adopts normalizing flow to learn balanced binary codes for the items.

**Strengths:**

1.	The ideas are presented clearly.
2.	The limitations are discussed.

**Weaknesses:**

1.	The methodology is outdated. In the field of vector search, the performance of LSH has been overwhelmed by vector quantization [1] and proximity graph [2] techniques by orders of magnitude in recent years. Pls check the two seed papers and the numerous papers that cite them. Although this fact may not be well known in the field of audio retrieval, I wonder a whether a simple adaption of proximity graph outperforms SOTA LSH methods for audio retrieval. The authors may consider switch to the latest techniques to vector search. Note that, vector quantization also has variants that use end-to-end training similar to this paper.

[1] Product Quantization for Nearest Neighbor Search
[2] Efficient and Robust Approximate Nearest Neighbor Search Using Hierarchical Navigable Small World Graphs (HNSW)

2.	Novelty is unclear. Section 3.1 uses contrastive learning to enlarge the distance gap between similar and dissimilar item pairs. This methodology is widely used in machine learning, and the authors should clarify the unique designs for audio retrieval. The core contribution seems to be using normalizing flow to ensure that the items are evenly distributed among the hash buckets. Again, the bucket balance constraint is widely recognized to be important and considered by many related works. Pls refer to the references in [1]. The authors should clarify the differences and advantages of normalizing flow over alternative bucket balance techniques.

[1] A Survey on Learning to Hash

3.	Experiments need to be improved. (1) Pls report the dataset statistics in a table, e.g., the number of items and the dimension. (2) When evaluating efficiency, pls use the query processing time. Currently, using the number of searched items cannot consider the costs of the encoding and hashing operations; plus that query processing time affects user experience more directly. (3) Pls use the time-recall curve to compare different methods, and an example can be found in the HNSW paper. Currently, Tabe 1 aligns neither the query time nor the recall of the methods, which makes it different to compare different methods. (4) Include the model training time of the proposed method.

**Questions:**

NA

---

> ### Author Response · Authors · 2023-11-17
>
> - "Novelty is unclear ... techniques"
>
> Indeed, we have adopted the encoder architecture proposed by Singh et al. Our primary innovation lies in the acceleration of hash-based indexing through the utilization of normalizing flow to learn balanced hash codes. While we acknowledge existing works, as mentioned in [1], that address the bit balance problem, our emphasis is on achieving balance at the bucket level on a large scale. It's worth noting that achieving bit balance ensures balanced marginals but doesn't necessarily extend to balancing the joint distribution encompassing all 2^K possible K-bit hash codes or buckets.
> Singh et al. proposed a solution for bucket balance using optimal transport formulation. However, we have identified scalability issues associated with their approach, primarily stemming from extensive computational and memory overhead. In response to this limitation, we advocate for the adoption of the normalizing flow model in our work. This model not only ensures the learning of balanced hash codes but does so in a more efficient manner compared to the optimal transport-based method. Importantly, our approach allows for achieving balance across hash buckets on a large scale, leading to a reduction in overall candidate evaluations during the search for the best match.
>
>
> - "Experiments need ... method"
>
> We provided a detailed description of the FMA dataset used for the experiments in Section 4.1. Due to ICLR constraints on paper length, we could not add comprehensive details about the dataset in the main paper, but we also provide relevant details about our system and the data dimension in the Appendix.
>
>
> Thanks for raising this interesting point. We chose to use the “number of searched items” as a raw metric for evaluating search algorithm efficiency. We opted for this metric due to the extensive optimizations present in open-source implementations such as faiss.ai, making a fair comparison in terms of query processing time challenging.
>
>
> Our primary contribution lies in the enhancement of hash-based indexing efficiency. We maintained similar recall rates as outlined by Singh et al.(TE+OT) demonstrating the speed-up achieved by our method (TE+NF) to attain comparable recall rates. Furthermore, our results highlight that our approach surpasses TE+LSH and TE+OT in terms of recall rates with fewer candidates, particularly for longer queries and across various distortion levels.
>
>
> The training duration of our model, encompassing both encoders and normalizing flow, requires approximately 5 days on a single A100 GPU.

---

> > ### Comment · Reviewer_q2gt · 2023-11-22
> > **After response**
> >
> > I have read the author response and decided to keep the original rating.

---

### Official Review · Reviewer_p7bp · 2023-10-31

**Soundness:** 3 good
**Presentation:** 4 excellent
**Contribution:** 3 good
**Rating:** 5
**Confidence:** 4

**Summary:**

The paper proposes a normalized flow-based hashing technique, which aims at providing uniformly distributed hash codes for a more efficient retrieval performance. Compared to an unstructured (unsupervised) hashing approach, their approach appears to be more beneficial as the search process is more streamlined. For the audio retrieval application, they proposed a three-step hashing process, where the first two are dedicated to learning representations from the sequence and then reduce to the dimensionality, while the third flow model makes sure that the final hash codes follow the organized bimodal distribution, thus making quantization easy. The experimental results show improvement although they are not significant.

**Strengths:**

- The paper is well-organized and easy to follow. The sections are clearly defined, and the math notations are clear, too.

- The proposed model consists of three different modules. It's clear that each of these modules has its own purpose. The combination makes sense, too.

- The employment of the normalization flow, which is the main contribution of this work, is a convincing choice and reasonable.

**Weaknesses:**

- The proposed method has other potential relevance to the hashing literature, which the paper doesn't discuss. For example, semantic hashing and LSH have this property that similar examples (either perceptually or from the application's perspective) tend to collide more often. In some audio retrieval tasks, this property might not be necessary, but it must be nice if the paper discusses this aspect, as to whether the proposed model results in these semantically relevant codes.

- The main weakness of the experimental results is, actually, the misrepresentation of the bold numbers in Table 1. It seems that the authors use bold characters to represent the best-performing model for each configuration, and the proposed model was chosen oftentimes, misleading the readers. However, with a careful examination, I was able to see that TE+LSH is the winner many times. It seems that TE+NF still slightly outperforms other methods in the noise-only case, but in other noise+reverb or reverb-only cases, TE+LSH is the clear winner. I believe that this is a critical mistake and the discussions and conclusion should be fixed accordingly.

**Questions:**

- In audio retrieval, for example, there can be semantically relevant examples (e.g., cover songs), that could share similar hash codes for a different applicational advantage. It appears that the proposed method relies heavily on the "exact match" scenario, which is also a legitimate application. Any additional explanation on this issue? This is also relevant to my other point about semantic hashing.

- The NF results are regularized to be a bimodal normal distribution. During training, these distributions are fixed with pre-defined mean and variances. Any chance this rigid definition could harm the hashing performance?

---

> ### Author Response · Authors · 2023-11-16
>
> Dear Reviewer,
>
> Thank you for your thoughtful review and valuable feedback on our paper. We appreciate the attention to detail and the insightful comments provided.
>
> Q1: "In audio retrieval ... semantic hashing"
> A1: We acknowledge the potential for scenarios where semantically related examples, like cover songs, might share similar hash codes for different applicational advantages.
> In our prior analysis, we particularly listened to the music mapped to the same hash bucket; however, we did not identify noteworthy patterns or findings to discuss in the paper. In response to your query, we further investigated if the same genre songs share similar hash codes, but our analysis did not reveal any significant patterns.
>
> While our method indeed relies on the "exact match" scenario, we understand the importance of addressing potential scenarios involving semantically relevant examples. We want to emphasize that our focus remains on the strengths and applicational advantages of the proposed method within the defined scope of exact matching. However, we acknowledge the relevance of your point and will consider exploring this aspect more thoroughly in future work.
>
>
>
> Q2: "The NF results ... performance?"
>
> A2:  We conducted comprehensive experiments to assess the influence of different means and variances on the overall indexing performance, specifically focusing on the recall rate and the number of candidate evaluations required to identify the best match. In our initial experiment, we varied the distance between the two modes by adjusting the means while maintaining a unit variance for each mode. Notably, as the distance between the two modes increased, we observed a decline in the recall rate. Conversely, reducing the distance between modes led to an uneven distribution of samples across hash buckets, resulting in increased candidate evaluations.
> In our understanding, increasing the distance between modes may pose a challenge for the model during training to map both x and x+ to the same mode if they lie in different modes. Furthermore, we also tried training the normalizing flow model with means and variances as learnable parameters. Through this approach, we observed the final learned means for both modes to converge around ±1.7 with a variance of 0.95. Therefore, we chose to have fixed means and variances for the two modes.
>
> Weakness: "The main weakness ... fixed accordingly"
>
> Response: In Tables 1 and 2, our primary objective is to demonstrate the efficacy of the two key components in our system: the transformer encoder for robust representation learning and the balanced buckets approach for optimized search efficiency. We show the effectiveness of our balanced bucket approach (NF) compared to OT and LSH while keeping the Transformer encoding fixed (TE+LSH vs TE+OT vs TE+NF). We show that our approach evaluated 3x and 1.6x less number of candidates on average than LSH and OT, respectively, to achieve similar recall rates across different distortion levels. We highlight our results only when the recall rate drop is marginal, with an accuracy gap of less than 1.5%, or when our approach surpasses the recall rates achieved by OT and LSH.
>
>
> Indeed, TE+LSH performs well compared to our TE+NF in noise+reverb and reverb environments for 1s queries but at the expense of more comparisons. Furthermore, our method (TE+NF) achieves similar or even surpasses for longer queries (2s, 3s, 5s) with a lesser number of candidate evaluations than LSH and OT, indicating the effectiveness of our subsequence search method (detailed in Section 3.4).
>
> Note that the main contribution of our paper lies in making hash-based indexing more efficient by reducing search candidates.

---

> > ### Comment · Reviewer_p7bp · 2023-11-22
> > **Confirm**
> >
> > Author response clarifies the contribution but it’s still limited to a certain scenario. Major performance issue is not addressed.

---

### Official Review · Reviewer_AxNq · 2023-11-01

**Soundness:** 3 good
**Presentation:** 2 fair
**Contribution:** 2 fair
**Rating:** 3
**Confidence:** 4

**Summary:**

This paper introduces FlowHash, a novel hashing scheme for solving the audio fingerprinting task.
They utilize the normalizing flows within the pre-trained Transformer-based encoder to obtain balanced K-bit hash codes, allowing efficient retrieval of the audio content.
Moreover, they incorporate a cross-entropy-based regularization term to achieve robust hash codes.
FlowHash proves to be an effective technique for reducing retrieval time and enhancing robustness against certain levels of noise and reverberation.

**Strengths:**

S1. **Novel Application of Normalizing Flows:**
It is interesting to leverage normalizing flows to achieve balanced hash codes.

S2. **Robust Hash Codes**:
They utilize the BCE loss as a regularization term to enhance the robustness of hash codes and perform experiments to validate this claim.

S3. **Comprehensive Experiments:**
They conducted a series of experiments that demonstrated the efficiency and robustness of the proposed method across various benchmarks.

**Weaknesses:**

W1. **Limited Novelty:**
The architecture and approach presented in this work bear strong resemblances to the methods proposed by Singh et al. (2023). Specifically, both papers leverage a Transformer-based encoder, which has already been extensively explored in the context of the prior work. The main extensions in this work over Singh et al. (2023) are the use of normalizing flows and the incorporation of the BCE loss. While these are indeed differences, since they are commonly used in other areas such as CV and NLP, their introduction might not be substantial enough to be considered groundbreaking.

W2. **Marginal Improvement over Accuracy:**
The paper highlights the capabilities of FlowHash in reducing retrieval candidates and enhancing robustness. These are undoubtedly important aspects in many practical applications. However, a critical metric for many machine learning tasks, particularly in the retrieval task, is accuracy. Based on the presented experimental results, it appears that the accuracy gains from using FlowHash are marginal. In some cases, e.g., from Tables 1 and 2, there might even be a trade-off where accuracy is sacrificed.

W3. **Extended Experiments:**
It might contain bias and becomes less convincing to conduct the experiments through a single dataset.
Moreover, while the paper emphasizes reduced memory overhead, it doesn't delve deep into the space overhead introduced by the hashing mechanism.

W4: **Incorrect Highlighted Results:**
In Tables 1 and 2, the authors have used bold fonts to signify the best results across different query lengths and varying levels of Noise and Reverberation. However, upon close inspection, there appear to be inconsistencies in the highlighted results, with several best values not being bolded correctly. Such incorrectly highlighted results can lead readers to draw wrong conclusions about the performance of the proposed method or its competitors.

**Questions:**

Regarding W3:

Q1: Can the authors include more real-world datasets for performance evaluation? It will be more convincing to justify the claims of robust and balanced hash codes with more datasets.

Q2: Can they report the memory overhead of different methods with the same recall rates? It will be beneficial to showcase the claim of memory reduction.

Regarding W4:

Q3: The authors should meticulously review Tables 1 and 2 to ensure that the best results are correctly highlighted. This might require re-checking the raw results to confirm the top performers.

**Details Of Ethics Concerns:**

No.

---

> ### Author Response · Authors · 2023-11-21
>
> Q1 and W3: Our decision to solely utilize the FMA dataset was based on its widespread adoption as a standard benchmark in recent literature for evaluating audio retrieval systems. However, we understand the importance of providing a more comprehensive analysis; therefore, we have extended our evaluation beyond the FMA database. We have now included results from experiments conducted on two additional databases, namely the LibriSpeech database and our in-house collection of broadcast content. These supplementary results can be found in the Appendix (A.1.1 and A.1.2), offering a broader perspective on the versatility and efficacy of our approach across different datasets.
>
>
> Q2: Thank you for raising the point. To provide further context on our indexing approach, our system encompasses three primary components: the fingerprints database, the hashtable index, and the metadata. Hereunder is the memory overhead occupied due to hash table construction for different approaches:
>
> LSH: 1.1 GB (10 hash tables)
>
> OT: 355 MB (single hash table)
>
> FlowHash: 676 MB (single hash table, but a single item is mapped to multiple hash buckets)
>
> To achieve a comparable recall rate accuracy, we need to evaluate more candidates in LSH and OT. While OT incurs a lower memory overhead than FlowHash, it introduces a computational overhead. This involves evaluating the distance between the query and all 2^K (typically K>=16) clusters to identify search candidates, thereby extending the retrieval process.
>
> Q3 and W2: Regarding your concern about the marginal accuracy gains presented in Tables 1 and 2, we would like to clarify that our primary focus is showcasing the search speed-up our method gains while attaining similar recall accuracy compared to baselines (OT and LSH). To ensure a fair comparison, we have adhered to the recall rates outlined by Singh et al. (2023). It is important to emphasize that the main contribution of our paper lies in making hash-based indexing more efficient by reducing the number of search candidates. We believe optimizing the indexing is pivotal for any retrieval system, particularly on a large scale. Our approach leads to a 3x acceleration in search speed while achieving similar or even surpasses recall accuracy across various distortion levels compared to LSH. Notably, our method exhibits scalability in contrast to OT.
>
>
> Q3 and W4: We highlight instances in the results where the recall drop is marginal, with an accuracy gap of less than 1.5%, or instances where our approach achieves a higher recall rate than the baselines while requiring 3x and 1.6x fewer candidate comparisons than LSH and OT, respectively.

---

> > ### Comment · Reviewer_AxNq · 2023-11-22
> > **Reply to Rebuttal**
> >
> > Thank you for your reply.
> >
> > I appreciate that the authors have included more datasets for performance evaluation and reported the hash table size. However, I still feel the contributions are insufficient as the novelty is limited, and the improvement is not very apparent. I tend to retain my score.

---

### Author Response · Authors · 2023-11-21
**Results clarification**

Dear Reviewers,

Thank you for your invaluable feedback and insightful comments regarding our paper. We acknowledge and apologize for any confusion that may have arisen from the presentation of results in Tables 1 and 2.
The primary focus of our paper is to underscore the effectiveness of balanced hash buckets in enhancing the speed of hash-based indexing methods. Specifically, we aim to showcase the speedup our proposed method (TE+NF) achieved in attaining recall rates comparable to the baseline methods (TE+OT and TE+LSH).

To address potential ambiguities, we have implemented changes in the presentation of our results. We now explicitly highlight results that exhibit the best recall rates, and additionally, we underline our findings only when the recall rate drop is less than 1.5% compared to the best result. This modification aims to clearly demonstrate that our method is 3x and 1.6x faster, with only a marginal drop (<1.5%) in some scenarios compared to LSH and OT, respectively. These modifications are intended to clarify and alleviate any confusion in interpreting our results.

Once again, we appreciate your constructive feedback and hope these adjustments enhance our paper's clarity.

---

### Meta-Review · Area_Chair_gAW9 · 2023-12-08

**Metareview:**

All reviewers recommend reject, mostly due to limited novelty of the approach and questions about whether its practical improvement is significant.

The authors emphasize that the relevance of the paper is not on an improved accuracy/recall but on improved search speedups compared to other approaches. This is good, but as noted by a reviewer, runtime search results depend on several things beyond the number of items searched. Also, the details of the implementation can easily alter the runtimes by a significant factor, which could make the claimed speedups disappear. It would be good to strengthen the experimental setup so as to be able to make an apples-to-apples comparison.

**Justification For Why Not Higher Score:**

See metareview

**Justification For Why Not Lower Score:**

N/A

---

### Decision · Program_Chairs · 2024-01-16

Reject